# Routine immunization intensification, vaccination campaigns, and measles transmission in Southern Nigeria

Niket Thakkar[1]*, Avuwa Joseph Oteri[2], Kevin A. McCarthy[1]

1 The Institute for Disease Modeling, Gates Foundation, Seattle, Washington, United States of America,
2 Nigeria Country Office, Gates Foundation, Abuja, Federal Capital Territory, Nigeria

* niket.thakkar@gatesfoundation.org

## Abstract

In this paper, we compare—in terms of their estimated effects on disease transmission—Southern Nigeria's large-scale measles vaccination campaigns since 2010 to a unique, more targeted routine immunization intensification that happened in 2019. A main focus of the discussion throughout is that quantifying intervention impact in real epidemiologies requires us to disentangle competing, dynamic sources of immunity, including unreported infection. To address this inference challenge, we create a collection of state-level, stochastic transmission models capable of estimating underlying measles susceptibility based on surveillance and survey data. Leveraging these models, we find that the 2019 intensification, despite being restricted in scale to children under 2 years-old, had an effect on transmission comparable to the region's larger vaccination campaigns targeting children up to 5. This implies that vaccines delivered in that effort were more than twice as likely to reach a susceptible child.

## Introduction

Access to measles vaccine is a fundamental global equity issue. While much of the world gets vaccinated as a routine health service, health systems in many countries are unstable and unreliable [1]. Millions of children are lost to routine systems every year, and with a lack of mechanisms for effective catch-up vaccination, they cannot get vaccine on time and are chronically left at risk.

Measles burden is also getting worse. There were an estimated 130000 measles deaths in 2022 [2], and cases reported to the WHO have increased since. The virus is highly contagious and particularly devastating for children under 5, with the risk of complications compounded by systemic issues like malnutrition or poor access to treatment [3]. Essentially all of this burden, which disproportionately affects the world's most vulnerable communities, is vaccine preventable [1], but global measles vaccination coverage has been essentially stagnant for nearly 15 years [4].

**Data availability statement:** Yes - all data are fully available without restriction; All relevant data is contained either within the paper or at the github repository: https://github.com/NThakkar-IDM/intensification.

**Funding:** The author(s) received no specific funding for this work.

**Competing interests:** The authors have declared that no competing interests exist.

Given this context, improving catch-up vaccination through routine systems and through concerted vaccination campaigns is a critical global health challenge. Making progress on it requires us to better understand what's worked well in the past and to translate those lessons into more effective implementation in the future. Research along these lines is evolving—studies have demonstrated the significance of campaign timing [5,6], outbreak response speed [7], and microplan quality [8]. But many more factors determine if a vaccine reaches a child before measles does.

This paper contributes to this broader effort. Specifically, we consider catch-up vaccination in Southern Nigeria, where immunity gaps have generally been addressed with large-scale vaccination campaigns targeting all children from 9 months to 5 years old. Five of these efforts have been undertaken since 2010, an example of a real-world implementation of the recommended catch-up frequency [9,10], but in 2019, Nigerian health authorities implemented an intensification of routine immunization (IRI) as part of the introduction of a second measles dose (MCV2) into routine services. That unique effort targeted 9 month to 2 year olds only and, unlike the campaigns, offered the full set of routinely delivered vaccines.

Comparing Southern Nigeria's 2019 IRI to its campaigns gives us an opportunity to understand how changes in age targeting and cadence affect outcomes in a high-burden setting. However, estimating intervention impact in real epidemiologies can be challenging. While there are direct measures of implementation quality, most notably post-campaign coverage surveys that measure how comprehensively doses were distributed, comparing interventions to one-another requires us to place them into context with disease-derived immunity, changes in routine immunization coverage, demographic shifts, and other potentially confounding dynamics.

We approach this inference challenge by creating detailed, state-level stochastic-process models of measles transmission for all 17 Southern states. These models take as input a broad dataset, including case-based reporting, household survey, and catch-up implementation data, and then estimate consistent epidemiological quantities like population susceptibility, transmission seasonality, and the rate of under-reporting. Taken together, across state-level models, we find that the vaccines delivered in the 2019 IRI were more than twice as likely to immunize a susceptible child as the vaccines delivered in the campaigns since 2010.

This paper is organized as follows. In the first section, we describe the input data and transmission model construction for Lagos State, which serves as a representative example. We show that the model is consistent with independent serological survey results and is predictive of future cases on a 3 year forecast horizon. Then, in the next section, we use the Lagos model to estimate the fraction of catch-up doses that immunized a susceptible child by event, which highlights the 2019 IRI as exceptional. Aggregating models across the South demonstrates that this result is robust. Finally, we conclude by discussing qualitative lessons for catch-up vaccine delivery, commenting on how this example contrasts with current guidance and offers empirical support for higher-frequency, narrower age-range efforts going forward.

## Modeling measles transmission—Lagos case study

### Epidemiological data

The data for Lagos State is broken into categories and visualized in Fig 1. As mentioned, inputs span multiple sources, but the primary sources are Nigeria's case-based surveillance system from 2009 through 2023, and household surveys, specifically the Demographics and Health Surveys (DHSs) and the Multiple Indicator Cluster Surveys (MICSs), from 2008 to 2021 [11–16].

Information from the case-based surveillance system is shown in red (top row). Measles cases that are lab confirmed, epidemiologically linked, or have a high estimated probability of confirmation based on individual age, self-reported vaccine history, and symptom-onset time (Appendix A) are aggregated daily. Lagos State's incidence curve suggests a

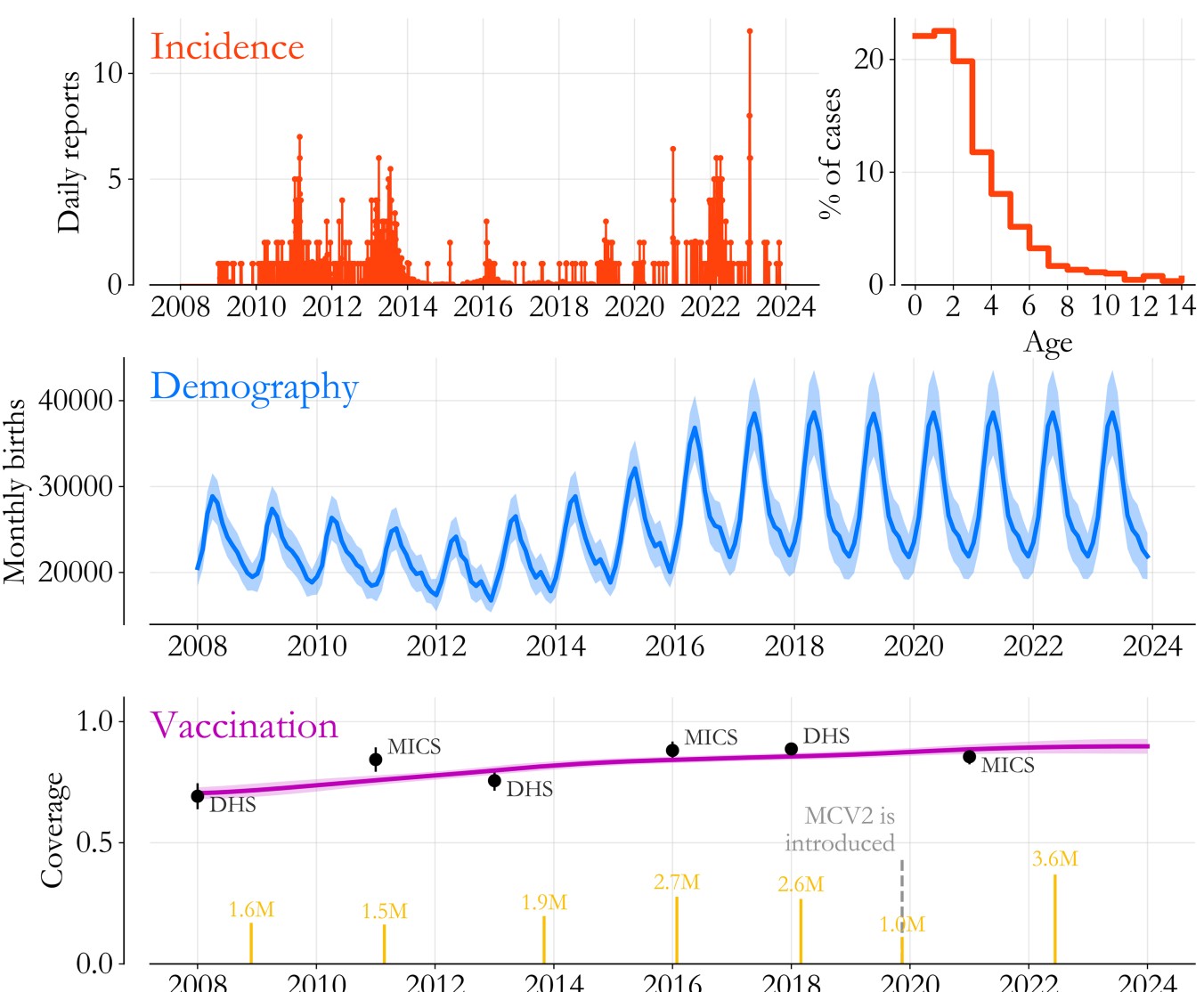

**Fig 1**. **Epidemiological data for Lagos State.** (Top) Both the time series of measles cases (red bars) and associated age-at-symptom onset data (red steps) come from Nigeria's case-based surveillance system. (Middle) We estimate the number of children born every month (blue, with 95% interval shaded) based on household survey data from 2008 to 2021. (Bottom) MCV1 coverage (purple, 95% interval shaded) is also based on the surveys, while catch-up event size (yellow bars) is estimated based on data from the WHO.

period of increased control after the 2013 vaccination campaign, and then a resurgence starting in 2021, in parallel with much of the world after Covid-19 disruptions to immunization programs [17]. Aggregated over time, the age distribution of cases (red steps) peaks for 1 to 2 year-olds, with a steady exponential decline for older ages, characteristic of a high-transmission-rate setting with initial maternal protection [18].

In Fig 1's middle panel, Lagos State's monthly births are estimated based on survey data. Specifically, we use a regression with post-stratification approach to estimate yearly births, and then we allocate them to months using an empirical seasonality profile based on surveyed birth dates. The average yearly birthrate associated with Fig 1 is roughly 28 births per 1000 population, within the Poisson interval of other estimates for urban Nigeria [13]. Sub-annually however, birth seasonality is a prominent effect across states, and in Lagos, we find that fluctuations reach nearly 30%, an estimate well within the range of country-level figures across Sub-Saharan Africa [19].

Finally, in Fig 1's bottom panel, we visualize the components of vaccine delivery. Coverage amongst 12 to 23 month-olds (purple), estimated with a similar regression and post stratification approach, is indicative of first dose routine coverage (MCV1) and shows steady increases from 2008 to 2024. Official estimates from individual surveys (black dots) give some modest validation of the statistical interpolation. Meanwhile, vaccination campaigns are indicated in yellow, with lines proportional to the number of doses delivered according to the WHO [4]. MCV2 introduction and the accompanying IRI are also indicated, with significantly fewer MCV doses delivered in that more targeted catch-up effort (Appendix B).

As a whole, Fig 1 highlights issues that motivate the need for modeling. For example, we estimate that roughly 30 thousand kids are born in Lagos State every month, and around 6 thousand of them are missed by routine MCV1. For the 54 thousand missed children born in the 9 months preceding a vaccination campaign, the age distribution of cases suggests that they experience nearly half of their infection risk before their next catch-up opportunity. Even in an extreme hypothetical where lifetime risk is determined by a 1% vaccine failure rate [20], we would still expect a few hundred cases per year from just one of these cohorts. But since 2009, Lagos State has reported less than 1200 cases in total, around 75 per year. Reconciling these apparent discrepancies, as well as estimating the impact of catch-ups on immunity, requires us to put the pieces together carefully.

## Model construction

Transmission models offer a guide to resolving the measurements in Fig 1 into a consistent epidemiological process. For our purposes here, we follow the approach in [21] closely—mathematical details can be found in Apps. C and D—and we assume that population can be compartmentalized into susceptible, infectious, and recovered states on a semi-monthly timescale [22,23]. Births, averaged to the same timescale, enter the susceptible population but are removed in proportion to routine MCV1 and MCV2 coverage after 9 and 15 months respectively. Those remaining are otherwise at risk of getting measles through interactions with infectious individuals, and new infections transition to being infectious within a semi-month before being removed from the system as well. Finally, during the six catch-up vaccination events from 2009 onward, susceptible individuals have additional opportunities to be immunized.

The model is fit to the incidence time series in Fig 1 conditional on the age, demographic, and vaccination-related information. We assume that the fraction of susceptible-infectious pairs leading to new infections is log-normally distributed with a seasonally varying mean [22] which we correlate across the states in Nigeria's geopolitical zones (collections of 5 or 6 contiguous states). Further, we assume infections are under-reported at an unknown, dynamic rate bounded by the expected annual burden consistent with the age distribution of cases in a survival analysis estimating the proportion of birth-cohorts left exposed across vaccination opportunities. Catch-up vaccination events are defined in the model by an efficacy parameter which represents the fraction of delivered vaccines that successfully immunized susceptible individuals.

The application of these ideas to Lagos State is visualized in Fig 2. In the top panel, the model (grey, 95% interval shaded) fits reported cases (black dots) closely with an underlying population prevalence (orange) reaching 1 per 1000 during outbreaks, similar to figures in other settings [22]. The transmission process is driven by underlying susceptibility (blue), which grows steadily as children are missed by routine immunization but falls precipitously at each of the 6 catch-up events (yellow bars). The overall level of susceptibility, roughly 6% of the population, is in very good agreement with the level measured by serological survey in 2018 ([24], grey bar), but with considerably higher confidence. Moreover, as we expected based on Fig 1, catch-up vaccination efforts from 2014 to 2020 closed large immunity gaps and controlled burden until late 2021.

In relating population prevalence to cases, we estimate that 0.24% (0.09 to 0.4% 95% confidence interval) of infections are reported to the surveillance system on average, with volatility in time (green) clearly proportional to, and maybe even a leading indicator of, the degree of burden control. This estimate is strikingly low, implying that each observed case in Lagos State represents roughly 400 infections. It's worth considering potential sources of error.

The reporting rate estimate would be biased low if for some reason the model's transmission process is too fast. One plausible issue is age-dependence in surveillance rates—maybe older individuals get measles but systemically are not reported?—and this biases the estimate of expected annual burden. However, the serological survey rules this possibility out. Survey estimates, which climb to near total immunity by age 5 in Southern Nigeria, suggest that the distribution in Fig 1 is representative, and more generally that large pockets of susceptible individuals cannot go years without encountering measles.

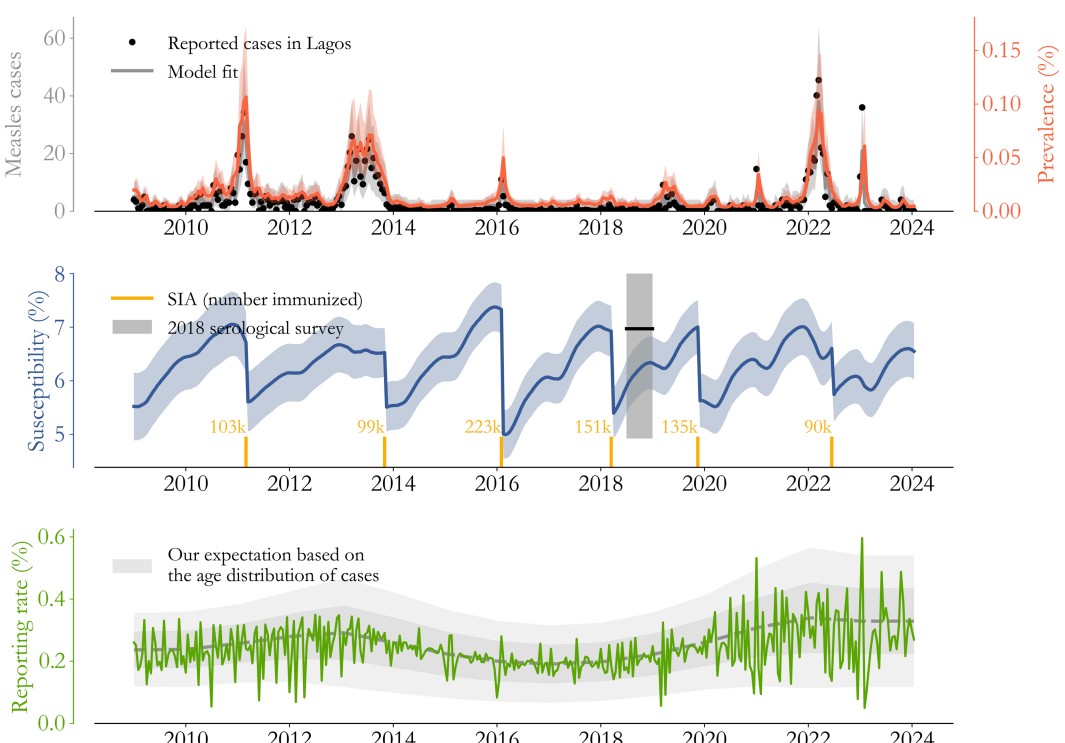

**Fig 2**. **Measles transmission in Lagos State.** (Top) The model (grey, 95% interval shaded) captures changes in cases (black dots) including the transitions to and from control, seen more clearly in terms of underlying prevalence (orange). (Middle) Model-based estimates of susceptibility (blue) rise as children are missed by routine immunization and fall during catch-up events, with an overall level corroborated by the serological survey in 2018 (black bar). (Bottom) The implied probability that infections are reported as cases (green) is low, dynamic, and volatile.

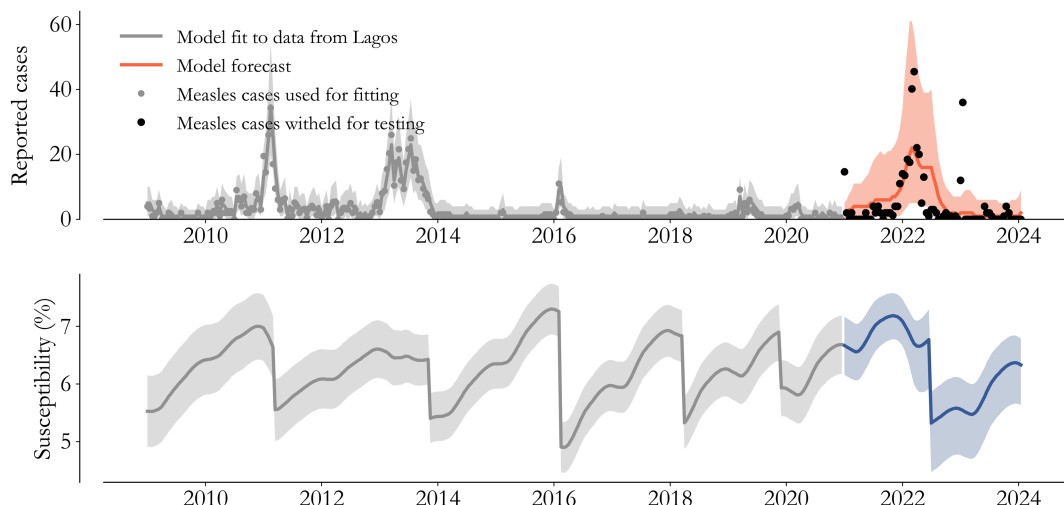

Other mechanisms that may slow transmission rates like network effects and age-dependence in contact structures are also bounded by the same argument. So while it's certainly true that these effects exist in Lagos State, it's implausible for them to change the order of magnitude of the reporting rate without unreasonably accumulating susceptibility. Surveillance issues in high-burden settings, like Nigeria and Pakistan [6], seem to warrant further study.

## Model validation

Stricter validation tests, on top of the fit to cases and the agreement with the serological survey, help us build confidence in the modeling approach and underlying inferences. Forecast accuracy is one such test, which tells us if the model's biological mechanisms and assessments of past catch-up activities are robust enough to anticipate burden.

We test the Lagos State model in Fig 3. The approach described in the previous section is applied to the data through the end of 2020 (grey), and the model is extrapolated until 2024 (color). This 3 year horizon was chosen for it's operational relevance, since catch-up efforts are generally separated by 1 to 3 years, and forecasts on that time-scale facilitate implementation.

With that operational context in mind, we assume that the timing of the 2022 vaccination campaign is known, as if it's being planned in early 2021, but the impact on immunity must be extrapolated. To do so, we take the average of past vaccination campaign impacts on susceptibility as the expected effect of the 2022 campaign. Furthermore, we extrapolate the reporting rate to be fixed at the average level estimated by the model at the end of 2020.

The black dots in Fig 3 are the cases actually observed in Lagos State from 2021 to 2024. The model-based forecast clearly captures the structure of the 2022 outbreak, it's timing, and the rate of it's decline due to the 2022 campaign. The 50% interval captures 49% of the data and the 95% interval captures 89% of the data. In the supplementary information, we show that this type of performance is typical of the models across the 17 states.

It's notable that, given cases alone, the rise in 2022 would seem historically unprecedented on the backdrop of control since 2014. Said differently, we would not expect a purely autoregressive model to capture the resurgence. Our approach clearly benefits from mechanistically relating demographic and vaccination information to outcomes, and this test helps us better understand those relationships.

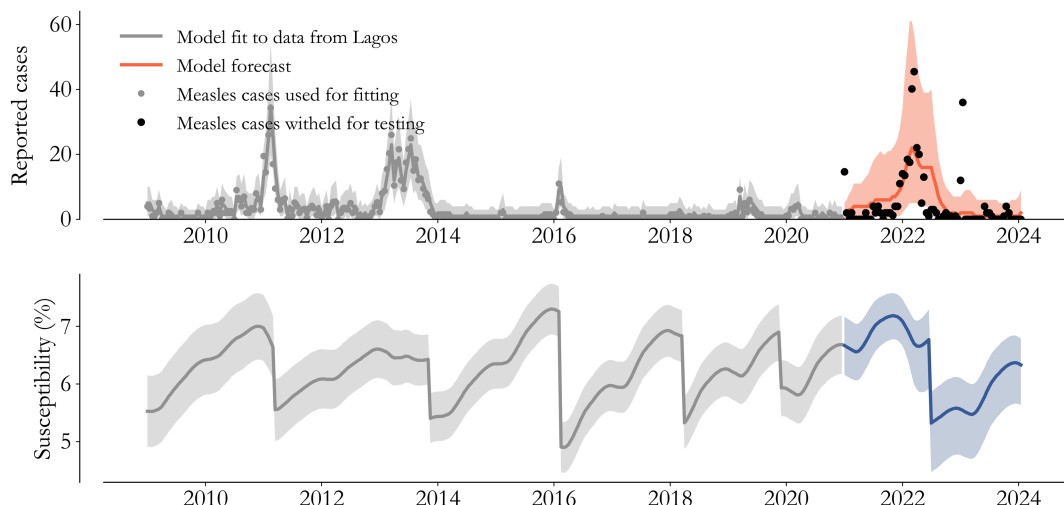

**Fig 3**. **Forecast testing.** Withholding data from 2021 onward (black dots) allows us to test model inferences (grey) and forecasts (orange). Forecasting necessarily involves predicting catch-up vaccination's impact on susceptibility (blue). The 2022 outbreak, a predictable result of a campaign delay, indicates what burden the 2019 IRI averted.

For example, the demographic and vaccination information essentially determines the rate of susceptibility increase during the years of control. In that era, larger catch-up effects would have placed the model at lower susceptibility entering 2022, and forecasts would have mistimed the outbreak as a result. Meanwhile, if catch-up effects were smaller, events like the 2022 outbreak would have been observed from 2014 to 2021. In the context of a transmission model, these features of the data constrain unknowns and lead to relatively high-confidence inferences.

## Comparing vaccine delivery modes

With the 17 state-level models in hand, we can return to our scientific and programmatic goals. We want to compare the 2019 IRI to the 5 vaccination campaigns.

Model-based estimates of per-dose efficacy offer a direct, quantitative measure of intervention impact that accounts for epidemiological context. More specifically, associated with each state's model is an approximate, joint posterior distribution over the parameters (Appendix D), which we can use to evaluate posterior profiles over each catch-up event's efficacy. This is visualized for Lagos in Fig 4, and the 2019 IRI (green) clearly stands out amongst the 6 catch-up efforts.

Relative to the campaigns (blue), there are two key differences. First, the posterior mode is markedly higher for the IRI - we estimate that 12.7% of IRI vaccines immunized a susceptible child, compared to just over 5.5% averaged across the campaigns. Second, uncertainty for the IRI is higher, a direct consequence of it's smaller size. This can also be seen by looking across the campaign distributions, and noting that the width is correlated to the number of doses delivered.

These differences are intriguing but should be interpreted carefully. An important limitation of the profiles, which hold all but the efficacy of interest constant, is that they can mask correlation between the interventions. For example, Lagos' 2018 campaign, which leveraged a variety of implementation innovations [25,26], had exceptionally high surveyed coverage [27]. This type of detail isn't directly incorporated into the model, and if the point estimate for the 2018 campaign is biased low, it's possible for the 2019 IRI profile to be biased high in a negative correlation that's not being explored. One potential issue is that there maybe other context specific correlations we're unaware of.

With that in mind, repeating this exercise across Southern states demonstrates that our findings are robust across state-specific implementation details. This is done in Fig 5. In the scatter plot, doses delivered are plotted against

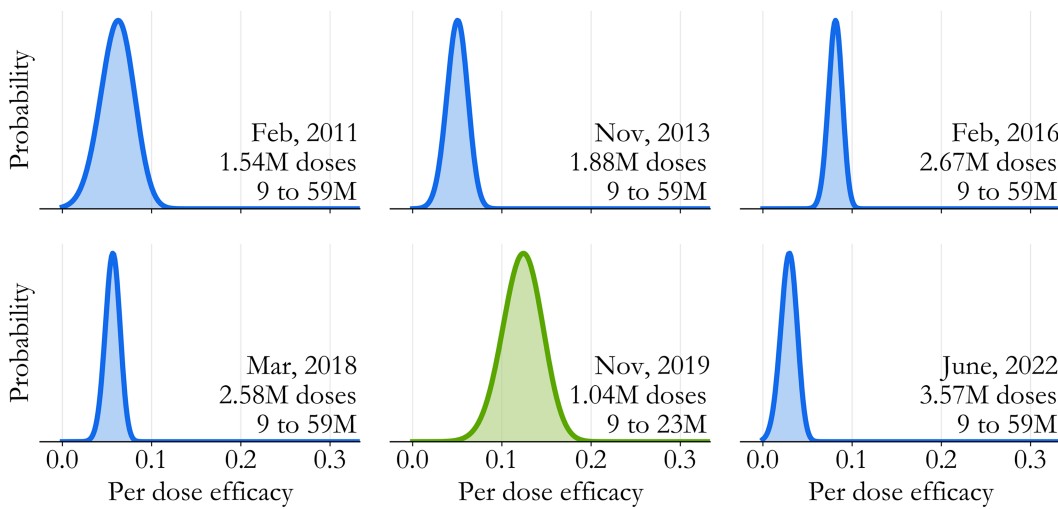

**Fig 4**. **Catch-up efficacy in Lagos State.** Transmission model-based posterior profiles can be computed for each catch-up event, essentially quantifying in terms of a probability how quickly the agreement with data changes as catch-up vaccines become more likely to immunize a child. The 2019 IRI (green) stands out amongst the campaigns (blue) even though the model lacks any age structure.

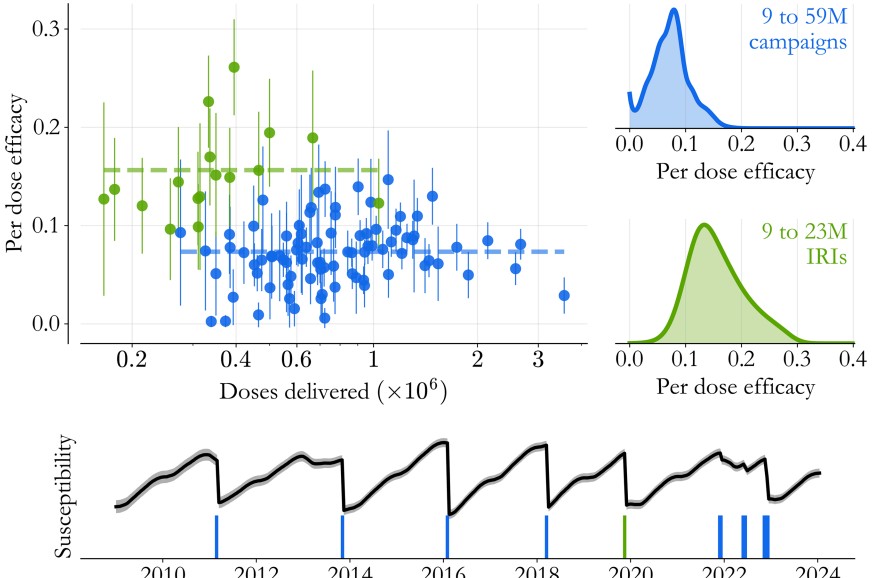

**Fig 5. Catch-up vaccination across the South.** State/catch-up event pairs (dots, 2 standard deviation error bars) across all 17 states maintain the distinction between the IRI (green) and the 5 campaigns (blue), across a wide range of event sizes. (Lower panel) Regional susceptibility (black) helps contextualize the catch-up event timeline (bars). The 2019 IRI (green) was less than 2 years after the 2018 campaign, in contrast to other, longer, inter-event times. Meanwhile, the events from 2021 onward were asynchronous across states.

estimated efficacy (2 standard deviation error bars) for individual state-catch-up event pairs, with the IRI highlighted in green. Averaged across catch-up event types (dashed lines), we find that 15.6% of IRI vaccines immunized a susceptible child as compared to 7.3% of campaign vaccines, a 2.13× difference. Posterior distributions mixed over the South show that uncertainty is higher for the IRI, as we saw in Lagos State alone, but that the distinction between the delivery modes remains significant.

Overall, it's striking that despite the model lacking any explicit age-structure, we can extract information relevant to age-targeting through principled inference, particularly by translating age-at-infection information into constraints on unknowns. As an estimation tool, the model presented here is versatile, parameterized in seconds on a laptop, and has an approachable data footprint. It offers a general, complementary approach to more conventional survey-based methods of evaluating interventions.

## Balancing drivers of efficacy

In retrospect, based on the distribution in Fig 1, we already expected susceptibility to decline with age, and it's not so surprising that campaigns targeting younger children are more effective per dose. But the effect in Fig 5 is large. In fact, we see from Fig 1 that the IRI delivered roughly half the doses of the average campaign (~2.3M doses), so we might coarsely expect campaigns to deliver 50% of their vaccines to children above 2 across the South (see Appendix B). If all of those older children were deterministically immune, the ratio of IRI to campaign efficacy would be ~2, lower than what we're seeing even in this limiting case.

It seems like there's more going on. The catch-up activity timeline is contextualized in terms of Southern Nigeria's susceptibility (black) in Fig 5's lower panel. One other striking feature of the 2019 IRI is that it was less than 2 years after the March 2018 campaign. In contrast, inter-campaign times since 2011 sometimes exceeded 2 and a half years.

From an epidemiological stand-point, this few month difference can be significant. In particular, in Southern Nigeria, the high-transmission season (Appendix D) begins in November, and the 2013 and 2022 resurgences are characterized by

inter-event times that span multiple high transmission seasons, supporting broader evidence [5] that campaign timing is a significant driver of overall impact.

In other words, the 2019 IRI's benefits were two-fold. As an intervention it was tailored to the highest susceptibility population, and in being smaller scale, it was possible to implement before that population was overexposed to measles risk. Delays in the execution of planned campaigns, particularly in 2013 and 2022, exposed susceptible children to additional high-transmission seasons, deflating the average campaign efficacy overall. The 2019 IRI is made substantially more effective in comparison.

## Conclusion

The Southern Nigerian epidemiology helps us evaluate different catch-up vaccination approaches. As it stands, vaccination campaigns and periodic IRIs (PIRIs) are generally thought of as two distinct options for vaccine delivery, and high-burden settings rely almost exclusively on campaigns to fill immunity gaps. But the Southern Nigerian experience calls for more flexibility.

The estimates in this paper point to a reframing of catch-up vaccination decision-making, moving away from minimizing event frequency with large target populations towards a system capable of maximizing frequency with small target populations. While the former objective was built around the idea that large campaigns interrupt transmission [9,10], the latter is more adapted to filling immunity gaps left open by past efforts.

A key finding along these lines is that even with perfect coverage campaigns, ineligible newborns in Southern Nigeria quickly age into susceptibility and are left at risk systematically. The current guidance relies on transmission interruption to protect these kids between catch-up events, but that seems unrealistic in practice. In fact, looking at susceptibility over time in Southern Nigeria (Fig 5) and considering the ever-present risk of delays in campaign execution, the minimum frequency strategy puts the population in a dangerous situation every 2 years essentially by design. It's clear that catch-up vaccination's global objectives require thoughtful reevaluation.

Till then, what we can say concretely is that Southern Nigeria's 2019 IRI had an effect on measles transmission comparable to its large campaigns. There are many implementation details needed to fully understand this out-sized impact, but the IRI's timing and targeting certainly played a significant role, as both were better adapted to the epidemiology. This historical example suggests that catch-up vaccination strategies have room for context-specific improvement as well.

Looking ahead, evaluating improvement will require us to complement measures of comprehensiveness with estimates of epidemiological impact. Transmission models are necessary to do that, helping us translate the data we observe into the underlying processes we can't, like the changes in susceptibility in the wake of a vaccination effort. The principles we've used in this study are general in the sense that they can be applied across settings, and they represent a step towards including more evidence in intervention design.

## Acknowledgments

This paper benefited significantly from feedback from a large group of colleagues. Specifically, we want to thank Katie Maloney, Ana Leticia Nery, Yusuf Yusufari, Shamsudeen Sani, and Bayero University's group led by Muktar Gadanya for their critical readings and their suggestions for edits. We also want to thank our colleagues at Nigeria CDC, US CDC, and NPHCDA for their collaboration and data collection expertise, without which this work could not have been done.

## Appendix

### A Untested, isolated, clinically compatible cases

Surveillance in Nigeria follows WHO guidelines [28] in the sense that febrile-maculopapular rash cases with cough, coryza, or conjunctivitis are considered clinically compatible measles cases. Blood samples taken from those individuals are generally tested for anti-measles IgM antibody and are either lab confirmed or rejected. If a clinically compatible case

can be linked by investigation to a lab confirmed case, lab testing is skipped and the case is classified as epidemiologically linked.

In practice, for a variety of reasons, not all clinically compatible cases complete this process. The situation is visualized for Lagos State in Fig 6. In the top panel, clinically compatible cases that were never tested or investigated (light grey) are stacked on top of lab confirmed and linked cases (dark grey). In 2013, the stark transition from dark to light grey illustrates a typical issue: labs can run out of materials for testing, and lab activity can be temporarily interrupted.

If all untested clinically compatible cases were considered measles, we would erroneously believe that transmission rates spontaneously increased in 2013. But that's clearly not the case. Lab rejected cases (black dashed line) show that clinically compatible cases are often due to other viruses, generally more than half the time in Lagos State.

Accurate assessment of the quality of interventions requires us to classify these remaining clinically compatible cases. To do so, we assume that the probability of a positive test for a symptomatic individual, age $a$ and with $d$ reported vaccine doses, is

$$p(\text{IgM} + |a, d, t) = \sigma\left(\beta_0 + \beta_1 I_{d=1} + \beta_2 I_{d\geq 2} + \beta_3 I_{d=\varnothing} + \beta_4 I_{a>5} + \varepsilon_t\right),$$

where $\sigma$ is the logistic function, $I_c \in \{0, 1\}$ is an indicator of condition $c$, and $\varepsilon_t$ is a random process over time $t$ correlated on the 2 month timescale. Missing dose histories, indicated by $I_{d=\varnothing}$, are treated as a special case. The parameters $\beta_i$ for $i = 0, .., 4$ measure relative changes in odds and are estimated jointly with $\varepsilon_t$ in a logistic regression [29] over lab confirmed and rejected cases.

The outcome of this approach for Lagos State is visualized in color in Fig 6. In red, the expected number of measles cases gracefully interpolates the lab interruption in 2013, exposing the period of measles control from 2014 to 2020 discussed in the main text. While individual-level uncertainty is high, aggregated uncertainty (red shading) is barely visible. Meanwhile, in the lower panel, test positivity rates averaged over $a$ and $d$ (blue) are in keeping with observed confirmation

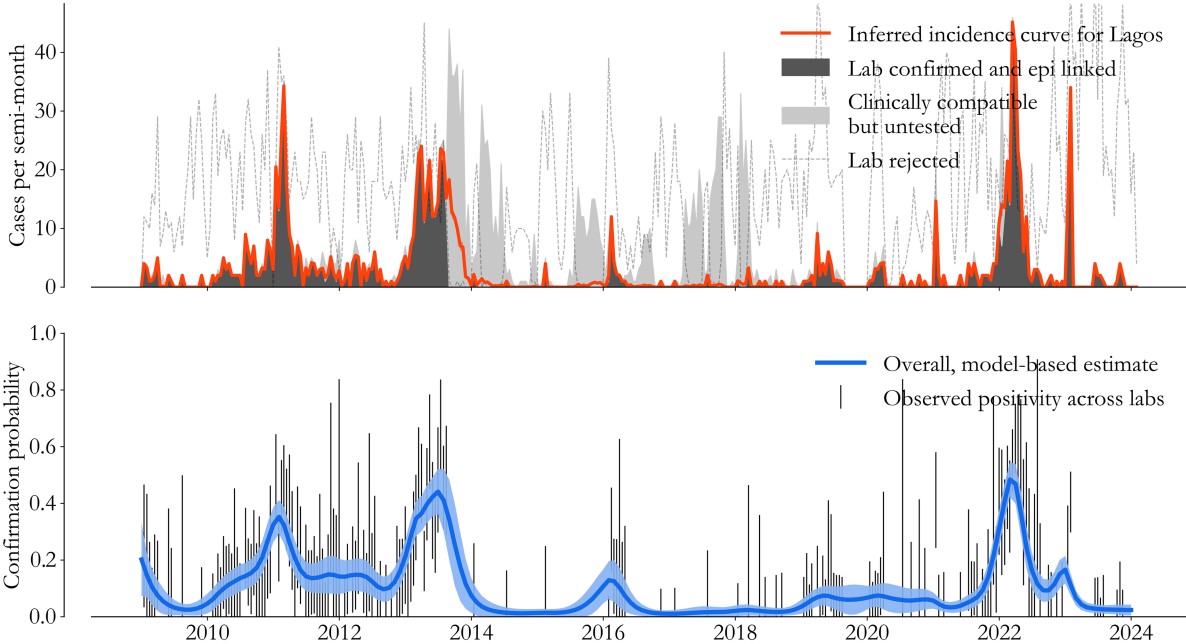

**Fig 6. Incidence curves.** (Top) We use a logistic regression approach to classify untested cases (light grey) and compile an incidence curve (red) for every state. (Bottom) Estimated confirmation probability (blue) follows lab activity closely (binomial error bars), exposing fast dynamics in the specificity of measles' clinical definition.

rates (binomial error bars), highlighting measles dynamics on a backdrop of other circulating pathogens. For the models throughout the text, this regression process is repeated for every state, and measles cases throughout this paper refer to the state's aggregated incidence curve.

**B Doses delivered in catch-up events**

The number of vaccines delivered in each catch-up activity is an important model input that comes with difficult-to-quantify structural uncertainties. We have partial information on catch-up event size at different spatial scales, and in general we're forced to make some assumptions on the size of the different events in each state.

For the campaigns, the WHO collects and publicly releases information on the number of doses aggregated over all the states participating [4]. To build state-level transmission models, we assume doses are distributed according to the distribution of children born 5 years prior to the campaign.

For the 2019 IRI, the WHO does not report the total number of doses delivered, and we're forced to incorporate additional implementation data. Specifically, we rely on ward-level (administrative level 3) aggregates of doses delivered by age in the 2022 campaign in Lagos, Gombe, and Ogun states, which we were able to obtain from the Nigerian Primary Health Care Development Agency.

The age distribution of doses delivered across all 726 wards (box plots), in comparison to what we would expect with a uniform age-distribution (blue), is visualized in Fig 7. Interestingly, younger children are over-represented on average, and 2 to 5 year-olds make up 49% of doses delivered, instead of the ∼ 70% we might guess for a 9 month to 5 year target. This seems intuitive, as older children may have already had opportunities for vaccine.

To specify the 2019 IRI in each state, we apply the average distribution in Fig 2 to each state's average 9 month to 5-year campaign size, and we collect the estimated number of vaccines delivered to children under 2. The full table of estimated vaccines delivered by state and event can be found in the repository associated with this paper.

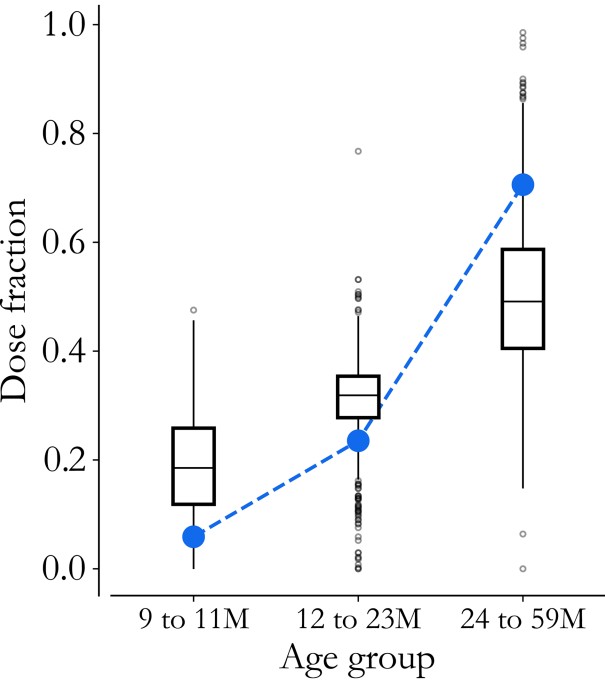

**Fig 7**. **Dose distributions.** Ward-level implementation data (box plots) from the 2022 vaccination campaign offers insight into the age distribution of children reached by catch-up events, which deviates from a binned uniform distribution (blue).

The above process is imperfect, and ideally we would have visibility into the full set of state-level delivery data. That said, in sensitivity testing, results throughout the paper were robust to reasonable changes in this upsampling approach. While the uncertainty in per-dose performance relies on this information, the uncertainty in relative efficacy is more stable. Moreover, in the transmission model, the key inference is the number of susceptible individuals immunized in each event, which is independent of doses delivered as long as that number is large. We are certainly in that limit since we expect most catch-up doses to be delivered to immune children, and we can safely compare catch-up activities as a result.

**C Survival-based priors**

Model-based inference follows the process in [30], generalized to incorporate vaccination and spatial correlation. A fundamental complexity in model-fitting is that, without some constraints, variation in cases can be explained either by a change in surveillance rates (e.g. catching more cases in a given time-period by chance) or by transmission dynamics (e.g. infectious people interacted with more susceptible people than usual). In reality, both of these effects happen simultaneously, and we need some mechanism for distinguishing them.

In the literature, this is often done by assuming reporting rates vary smoothly [22] or not at all [6,23]. We take a less restrictive approach here: We first calculate expected burden at an annual timescale, using the age-at-infection distribution to allocate birth-cohorts to infections over time (sometimes called a cohort model or an immunity profile [31]). Then, in comparing expectations to cases, we can estimate slow variation in the surveillance rate, which we use as an informative prior in the context of a more conventional, fast time-scale transmission model.

To illustrate the basic idea, consider a birth-cohort, size $B_b$, labeled by birth year $b$. If we assume that individuals in that cohort acquire measles immunity once and then have it for life, we can try to partition the cohort into groups associated with their immunity source, $s \in \{I, V_1, V_2, V_1^c, ..., V_M^c\}$, where $I$ is infection, $V_1$ and $V_2$ are the routine vaccines, and $V_i^c$ are catch-up vaccines associated with events $i = 1, ..., M$.

In this context, the age distribution of cases gives us insight into $p(a|s = I, b)$, the probability of being age $a$ given that an individual is infected and born in year $b$. The object of primary interest for us, however, is the joint distribution

$$p(a, s = I|b) = p(s = I|b)p(a|s = I, b),$$

where the prefactor represents the overall fraction of cohort $b$ destined to be infected. In other words, our goal is to calculate the fraction of each birth cohort left to infection across all ages, and then allocate them in time according to the age information associated with cases.

Since opportunities for vaccine-derived immunity are ordered, limited, and largely disjoint, we can make progress case-by-case. In general, each vaccination opportunity happens at a fixed time, $t_{s'}$, has a known probability of seroconversion, $\varepsilon(s')$, and an associated coverage, $C(s'|b)$. Then,

$$p(s = s'|b) = \varepsilon(s')C(s'|b)\left[1 - p(s = I|b)p(a \leq t_{s'} - b|s = I, b) - \sum_{s'' < s'} p(s = s''|b)\right],$$

where in the survival term $s'' < s'$ refers to the chronological ordering of vaccine opportunities and for $s = I$ we leverage the conditional age distribution. For a given cohort $b$, across $p(s = s'|b)$, these recursive equations can be evaluated at various $p(s = I|b)$. Noting that $\sum_s p(s|b) = 1$ is linear in $p(s = I|b)$, we evaluate the recursion at 2 points and then solve for the $p(s = I|b)$ that exactly balances the normalization condition for each cohort.

In application to Nigeria, we assume $\varepsilon(V_1) = 0.825$, $\varepsilon(V_i^c) = 0.9$, and $\varepsilon(V_2) = 0.95$ [20]. Coverage of MCV1 is estimated as described in the main text, and for MCV2 it's scaled for each state by the yearly ratio of national-level MCV2 coverage to MCV1, since we lack MCV2-related survey data. For the catch-up vaccines, we assume $C(V_i^c|b)$ is equal to the fraction of cohort $b$ expected to be within the age-range given the state's birth-seasonality profile, discounted by an assumed 90%

coverage amongst the target-population. Finally, we estimate $p(a|s = I, b)$ in a categorical regression against the age information associated with cases assuming the distribution varies smoothly in $a$ and $b$ and has negligible support for $a > 25$ years-old.

Given the distribution $p(a, s = I|b)$, expected annual burden is $E[I_t] = \sum_b B_b p(a = t - b, s = I|b)$. We can also calculate the expected initial susceptible population (those destined to be infected or vaccinated at $t \geq 2009$ from cohorts $b < 2009$) and associated variance in susceptibility across outbreaks in a similar way, assuming $p(a|s = I, b)$ is static for $b \leq 2002$ and interpolating MCV1 coverage from the introduction year (1978) to the earliest year we have data.

Application of this approach is visualized for Lagos State in Fig 8. Bars on the left indicate the fraction of each cohort with immunity across sources as of the start of 2024. Routine first-dose coverage (black) is the dominant contribution in Lagos, but catch-up vaccination events (blue shades by event) are critical. Fractions of cohorts missed by vaccination are highlighted in red, exposing periodic increases. The 2 to 3 year frequency of catch-up events leads to chronically missed populations that amount to thousands of infections.

Regression against cases (blue) is shown on the right, and while this simplistic model lacks the interactions needed to produce outbreaks, we capture the observed level of burden. The associated reporting rate (purple, 1 and 2 standard deviations shaded) rises in recent years, but with the level remaining low overall. In trying to understand the source of the rise, we can overlay the trend in rejected cases (black dots), which is indicative of overall surveillance activity. The agreement loosely corroborates our inference and suggests that Lagos is in a situation where increasing testing volume would increase measles reporting rates.

## D Building a complete transmission model

The approach in the previous section allows us to constrain the surveillance effects in established measles transmission models. But with the low reporting rates we've uncovered, we also expect case time series to be sparse, limiting the

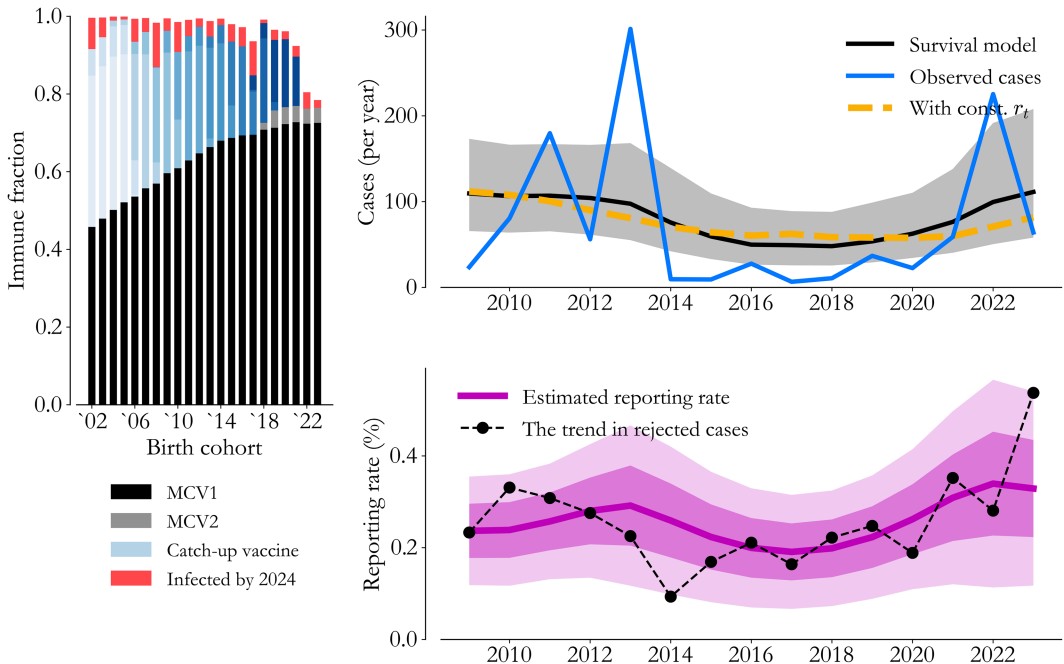

**Fig 8**. **Survival and immunity.** (Left) We leverage a survival analysis across vaccination opportunities (colors) to partition birth cohorts by immunity source. (Right) Infections across cohorts can be compared to observed cases (blue) to estimate annual-scale variation in reporting rates (purple).

quality and stability of inferences. With that in mind, in this section, we describe the model in detail, emphasizing the use of spatial correlation to stabilize seasonality estimates.

We assume susceptible individuals, $S_t$, and infectious individuals, $I_t$, stochastically interact with one-another in semi-monthly increments $t$, approximating the roughly 14 day exposure to rash-onset time associated with measles [22,23]. If infectious individuals are immune for life, considering first a population in isolation, we have

$$S_{t+1} = S_t + B_t - V_t - I_{t+1} \tag{1}$$

$$I_{t+1} = (\gamma \beta_t \varepsilon_t) S_t I_t^{\alpha} \tag{2}$$

where $\beta_t$ is an annually periodic transmission rate, $\varepsilon_t$ is a log-normally distributed volatility with variance $V[\varepsilon_t] = \sigma_\varepsilon^2$, and $\alpha \leq 1$ accounts for individuals capable of being infected without infecting others [22]. Susceptibility in the model accumulates with new births, $B_t$, and declines with both infections and with immunizing vaccines, $V_t$.

Routine immunization is the main contributor to $V_t$, based on coverage estimates, $C_t(V_1)$ and $C_t(V_2)$, applied to $B_t$ with a 9 and 15 month lag respectively. Otherwise, vaccines come from catch-up events $i = 1, ..., M$, which contribute terms of the form $\mu_i d_i$, an unknown per-dose efficacy scaling the assumed known number of doses delivered.

To stabilize inferences, consider a related model for a larger population, which we call the neighborhood of the process defined by Eqs 1 and 2. In the neighborhood, we have

$$S_{t+1}^N = S_t^N + B_t^N - V_t^N - I_{t+1}^N \tag{3}$$

$$I_{t+1}^N = (\beta_t \varepsilon_t^N) S_t^N I_t^N, \tag{4}$$

where neighborhood quantities are marked by an $N$ superscript and are defined as above. Note specifically that Eq 2 and Eq 4 share a seasonal transmission rate, $\beta_t$, with only a constant scale-factor $\gamma$ in the smaller population. While this relationship doesn't allow infection or susceptibility to move from the neighborhood to the population of interest, it allows us to more intentionally inform seasonal variation in transmission under the assumption that the drivers of seasonality are shared.

In application to Nigeria, we model transmission at the state-level with Eqs 1 and 2, and we consider the state's geopolitical zone (a collection of 5 or 6 states in Nigeria) to be the neighborhood. In each of the neighborhood's $Z$ states, we assume,

$$C_t \sim \text{Binomial}\,(I_t, r_t) \tag{5}$$

where $C_t$ are observed cases and $r_t$ is the semi-monthly reporting rate. This allows us to calculate statistics like $E[I_t | C_t, r_t]$, $E[S_0]$, and $V[S_0]$ for every state and then sum to the zone-level as needed.

To specify the model, we start with the neighborhood process. Consider the dataset $\mathbf{D} = \{C_t^i, B_t^i, C_t^i(V_1), C_t^i(V_2), d_1^i, ... d_{M^i}^i \mid i = 1, ... Z\}$ over the $Z$ relevant states in a given zone. If we assume the state of interest corresponds to $i = 1$, and we label the annually-aggregated dataset $\tilde{\mathbf{D}}$, then the joint inference distribution for the neighborhood parameters is

$$\begin{aligned}
p(\boldsymbol{\mu}^N, &S_0^N, r_t^2, ..., r_t^Z, I_t^N, \beta_t, \varepsilon_t^N | \mathbf{D}, \tilde{\mathbf{D}}) \\
&= p(\boldsymbol{\mu}^N | \mathbf{D}, \tilde{\mathbf{D}}) p(S_0^N, r_t^2, ..., r_t^Z | \boldsymbol{\mu}^N, \mathbf{D}, \tilde{\mathbf{D}}) \\
&\quad p(I_t^N | S_0^N, r_t^2, ..., r_t^Z, \boldsymbol{\mu}^N, \mathbf{D}, \tilde{\mathbf{D}}) p(\beta_t, \varepsilon_t^N | \boldsymbol{\mu}^N, S_0^N, r_t^2, ..., r_t^Z, I_t^N, \mathbf{D}, \tilde{\mathbf{D}}) \\
&\approx p(\boldsymbol{\mu}^N) p(S_0^N | \tilde{\mathbf{D}}) p(r_t^2, ..., r_t^Z | \tilde{\mathbf{D}}) p(I_t^N | \mathbf{D}, r_t^2, ..., r_t^Z) p(\beta_t, \varepsilon_t^N | \mathbf{D}, S_0, I_t^N, \boldsymbol{\mu}^N),
\end{aligned}$$

where $\mu^N$ is a vector containing every state's catch-up event efficacies. In the first line above we've chosen an exact hierarchical decomposition, and in the second line we've made some conditional independence assumptions. From left to right, along the lines of the logic in [30], the first term is a prior we use to enforce $0 \leq \mu^N \leq 1$ element-wise, the second and third terms correspond to the survival analysis of the previous section, the next term refers to the observation process in Eq 5, and the final term to the transmission process in Eq 4. For the neighborhood, we treat the observation process coarsely, assuming $r_t^i = \mathrm{E}[r_t^i | \tilde{\mathbf{D}}]$ deterministically which implies $I_t^N = \sum_{i=2}^{Z} \mathrm{E}[I_t^i | C_t^i, r_t^i]$. The final term is then a well-specified log-linear regression, and we can find the joint maximum posterior estimate with standard optimization methods.

With estimates of $S_t^N$ and $I_t^N$ in hand, we can approach Eqns. 1 and 2 in a similar fashion, with a joint distribution as above but conditional on the neighborhood states as well. This time, however, we allow $p(r_t | \tilde{\mathbf{D}})$, the survival prior, to have non-zero variance, and we only neglect the binomial variation in $p(I_t | \mathbf{D}, r_t)$, so that $I_t = \mathrm{E}[I_t | C_t, r_t]$ mixed over $p(r_t | \tilde{\mathbf{D}})$. The transmission regression now incorporates both models,

$$
\begin{aligned}
\ln I_{t+1} - \ln S_t &= \ln \gamma + \ln \beta_t + \alpha \ln I_t + \ln \varepsilon_t \\
\ln I_{t+1}^N - \ln S_t^N - \ln I_t^N &= \ln \beta_t + \ln \varepsilon_t^N,
\end{aligned}
\tag{6}
$$

with specified covariates and responses for $t = 1, .., T$. The negative log-posterior is, up to a constant,

$$
\mathcal{L}(\gamma, \beta_t, \varepsilon_t, \alpha, r_t, S_0, \mu_1, ..., \mu_M) = (24T - 1) \ln \hat{\sigma}_\varepsilon^2 + \frac{(S_0 - \mathrm{E}[S_0])^2}{2\mathrm{V}[S_0]} + \sum_t \frac{(r_t - \mathrm{E}[r_t])^2}{2\mathrm{V}[r_t]},
\tag{7}
$$

where $\hat{\sigma}_\varepsilon$ is the maximum-likelihood estimate of $\sigma_\varepsilon$ associated with the regression in Eq 6. Eq 7 emphasizes the balance between the slow and fast dynamics, with the first term capturing fast transmission and the final term incorporating the slow variation in reporting. We can minimize $\mathcal{L}$ to find the best fit model, differentiate it twice to estimate uncertainty, and calculate profiles in specific parameters like $\mu_i$ (see Table 1 for reference). In practice, we approach this optimization problem with the same algorithm as in [30] (see footnote 3 in that paper), and we sample the best-fit model to calculate forecasts.

Incorporating the neighborhood process gives us more informed seasonality profiles with higher certainty. For completeness, the seasonality profile for Lagos State is visualized in Fig 9. The periodic variation starts to rise in November, as mentioned in the main text, and there is up to $\sim 40\%$ variation from high to low season transmission. As a result, this parameter is both critical for forecast accuracy and highly relevant for catch-up vaccination implementation. Spatial correlation is therefore a necessary model feature to make estimates at this spatial resolution in a low reporting rate setting.

**Table 1**. **Reference table for inferred parameters.** The regression in Eq 7 across these parameters specifies the transmission and observation dynamics in a given state. With periodic $\beta_t$ and $\varepsilon_t$ defined by its constant variance, $\sigma_\varepsilon^2$, there are $28 + T + M$ unknowns in principle. The regularization of Appendix C is essential to make the problem well-defined.

| Parameter | Length | Description |
|---|---|---|
| $\beta_t$ | 24 | Annually-periodic transmission rate for both the state and neighborhood |
| $\gamma$ | 1 | Dimensionless scale-factor to adjust $\beta_t$ from neighborhood to state-level |
| $\varepsilon_t$ | 1 | Log-normal volatility in transmission |
| $\alpha$ | 1 | Exponent on $I_t$, the infectious population, with $\alpha \leq 1$, (see [22,30] for more details) |
| $r_t$ | $T$ | Reporting rate, the probability that infections are reported as cases at time $t$ |
| $S_0$ | 1 | The initial ($t = 0$) count of the susceptible population |
| $\mu_1, ..., \mu_M$ | $M$ | Intervention efficacies, the fraction of doses delivered that immunized a susceptible |

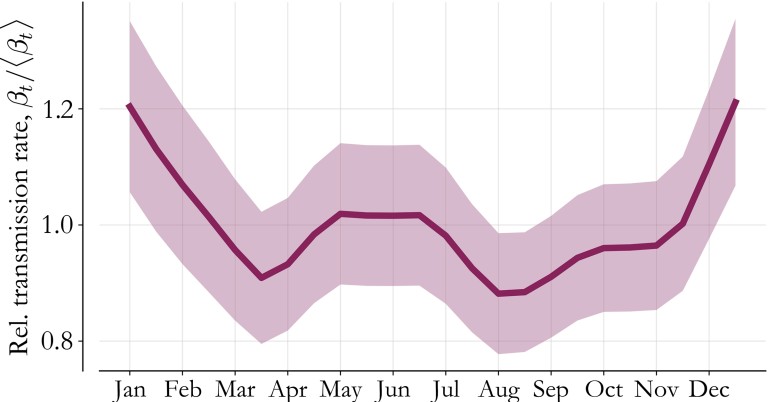

**Fig 9**. **Transmission seasonality.** Seasonality in Lagos State is estimated during model-fitting. The high-transmission season begins in November and continues through February.

## Author contributions

**Conceptualization:** Niket Thakkar, Avuwa Joseph Oteri, Kevin A. McCarthy.

**Data curation:** Niket Thakkar, Avuwa Joseph Oteri, Kevin A. McCarthy.

**Formal analysis:** Niket Thakkar.

**Investigation:** Niket Thakkar.

**Methodology:** Niket Thakkar, Kevin A. McCarthy.

**Software:** Niket Thakkar.

**Validation:** Niket Thakkar, Avuwa Joseph Oteri.

**Visualization:** Niket Thakkar.

**Writing – original draft:** Niket Thakkar, Avuwa Joseph Oteri, Kevin A. McCarthy.

**Writing – review & editing:** Niket Thakkar, Avuwa Joseph Oteri, Kevin A. McCarthy.

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
