## [Decision Letter · Decision Letter 0]

12 Nov 2025

PONE-D-25-54423Routine immunization intensification, vaccination campaigns, and measles transmission in Southern NigeriaPLOS ONE

Dear Dr. Thakkar,

Thank you for submitting your manuscript to PLOS ONE. After careful consideration, we feel that it has merit but does not fully meet PLOS ONE’s publication criteria as it currently stands. Therefore, we invite you to submit a revised version of the manuscript that addresses the points raised during the review process.

We look forward to receiving your revised manuscript.

Kind regards,

Sk Md Mamunur Rahman Malik

Academic Editor

PLOS ONE

Reviewers' comments:

Reviewer's Responses to Questions

**Comments to the Author**

1. Is the manuscript technically sound, and do the data support the conclusions?

Reviewer #1: Yes

Reviewer #2: Yes

Reviewer #3: Yes

Reviewer #4: Partly

2. Has the statistical analysis been performed appropriately and rigorously?

Reviewer #1: Yes

Reviewer #2: Yes

Reviewer #3: Yes

Reviewer #4: I Don't Know

3. Have the authors made all data underlying the findings in their manuscript fully available?

Reviewer #1: Yes

Reviewer #2: Yes

Reviewer #3: Yes

Reviewer #4: Yes

4. Is the manuscript presented in an intelligible fashion and written in standard English?

Reviewer #1: Yes

Reviewer #2: Yes

Reviewer #3: Yes

Reviewer #4: Yes

5. Review Comments to the Author

Reviewer #1: This is a well-executed and insightful study that makes a clear contribution to understanding measles transmission dynamics and vaccination impact in Southern Nigeria. The modeling framework is rigorous, the analysis is transparent, and the conclusions are well supported by the data. The comparison between large-scale campaigns and the targeted 2019 intensification provides valuable policy-relevant insight into optimizing immunization strategies. The manuscript is well written, logically structured, and requires no revisions. I recommend acceptance as is.

Reviewer #2: The authors investigate the impact of an intensified, targeted measles vaccination campaign that complemented existing supplemental immunization activities (SIAs) in reducing the susceptible population in Nigeria. Their objective is to evaluate whether these targeted campaign more effectively reach unvaccinated individuals who are typically missed by routine or prior SIA efforts.

To achieve this, the study integrates multiple data sources, including serosurveys, Multiple Indicator Cluster Surveys (MICS), Demographic and Health Surveys (DHS), and case reporting estimates, to parameterize a stochastic time-series susceptible–infected–recovered (TSIR) model. The model extends the classical TSIR framework originally introduced by Finkenstädt and Grenfell (2000) by incorporating time-varying reporting rates and demographic renewal through birth data, while drawing from nationally representative surveys, including the Nigeria Demographic and Health Surveys and Multiple Indicator Cluster Surveys to inform population and vaccination trends. This integration of demographic data and mechanistic modelling provides a data-driven reconstruction of susceptible dynamics, offering insights into how health-system improvements, campaign rollouts, and surveillance evolution may have shaped observed measles patterns. The same modelling framework was prior applied to measles dynamics in Pakistan and Somalia by the authors, where it successfully demonstrated the model’s ability to disentangle true transmission changes from variations in reporting.

In the present work, the authors adapt this framework to Nigeria to quantify the effectiveness of an intensified measles campaign, estimating the efficacy per vaccine dose. Despite delivering roughly half the total number of doses compared with the standard SIA, the intensified campaign achieved a higher probability of reaching previously missed susceptible children.

This finding offers valuable insight into measles control and elimination efforts in high-burden settings, particularly those where young children transition from maternal antibody protection to susceptibility before being reached by routine or catch-up immunization. The results carry clear policy relevance for optimizing campaign design and targeting in the final stages of measles eradication as well as insights in outbreaks forecasting.

It would be interesting to see the estimated averted case burden (similar to the authors’ Somalia paper). A broader audience, including policy makers, would benefit from seeing the trajectory of cases without the intervention compared with those averted by the campaign.

The following refinements would improve readability and accessibility for a broad epidemiologic audience:

Minor Comments

1. Model Construction:

1.1. Move the core model description and assumptions into the main Methods section rather than leaving them solely in the appendix.

1.2. Add a visual summary showing data inputs (births, MICS/DHS, campaigns) and model compartments and flows. This would significantly improve clarity for non-modelling readers.

1.3. Parameter table in the appendix:

Include a concise parameter summary table listing each symbol, its description, unit, and epidemiologic interpretation. This addition would enhance accessibility and reproducibility.

2. Figures and visual aids:

2.1. Label all multi-panel figures as (a), (b), (c) rather than “top,” “middle,” or “bottom” for consistent referencing.

2.2. When citing figures in the text, explicitly note which panel supports each statement (e.g., “Figure 2b shows…”).

2.3. Ensure that figure captions are self-contained, describing axes, color coding, and what each panel represents.

Reviewer #3: This manuscript is interesting and useful for operational vaccination activities. It is easy to read, despite a few typos (for example, the word "catagorical"). The figures are legible. The authors should specify the limitations of this study, and the model needs to be validated by analyzing data from several countries.

Reviewer #4: The manuscript was well prepared and technically sound too. The statistical analysis were performed rigorously. The authors too have tried to make the findings available on the manuscript and on GitHub. This manuscript was also written in standard English. However, the conclusion of the research work reflects that Southern Nigeria’s 2019 IRI had an

effect on measles transmission comparable to its large campaigns. The graphs shown for this work in all forecasting and susceptibility did not present adequate parallel comparison with 2019 and other years in order to support the conclusion. I have also found that this research work was part of a work submitted to an Australian National University

6. PLOS authors have the option to publish the peer review history of their article (what does this mean?). If published, this will include your full peer review and any attached files.

Reviewer #1: **Yes:** Md Mehedy Hasan Miraz

Reviewer #2: No

Reviewer #3: No

Reviewer #4: No

<qb-div data-qb-element="re-enable-flow" style="z-index: 2147483647; max-width: 1px; max-height: 1px; box-sizing: border-box; position: fixed; top: 10px; right: 10px;"><qb-div style="all: initial !important;"></qb-div></qb-div>

---

## [Author Response · Author response to Decision Letter 1]

3 Dec 2025

The response to reviewers is attached in the response to reviewers PDF.

---

## [Editor Report · Decision Letter 1]

30 Dec 2025

Routine immunization intensification, vaccination campaigns, and measles transmission in Southern Nigeria

PONE-D-25-54423R1

Dear Dr. Thakkar,

We’re pleased to inform you that your manuscript has been judged scientifically suitable for publication and will be formally accepted for publication once it meets all outstanding technical requirements.

Kind regards,

Sk Md Mamunur Rahman Malik

Academic Editor

PLOS One
---

## [Editor Report · Acceptance letter]

PONE-D-25-54423R1

PLOS One

Dear Dr. Thakkar,

I'm pleased to inform you that your manuscript has been deemed suitable for publication in PLOS One. Congratulations! Your manuscript is now being handed over to our production team.

Kind regards,

on behalf of

Dr. Sk Md Mamunur Rahman Malik

Academic Editor

PLOS One